# A GENERALISED INVERSE REINFORCEMENT LEARNING FRAMEWORK

## ABSTRACT

The global objective of inverse Reinforcement Learning (IRL) is to estimate the unknown cost function of some MDP based on observed trajectories generated by (approximate) optimal policies. The classical approach consists in tuning this cost function so that associated optimal trajectories (that minimise the cumulative discounted cost, i.e. the classical RL loss) are "similar" to the observed ones. Prior contributions focused on penalising degenerate solutions and improving algorithmic scalability. Quite orthogonally to them, we question the pertinence of characterising optimality with respect to the cumulative discounted cost as it induces an implicit bias against policies with longer mixing times. State of the art value based RL algorithms circumvent this issue by solving for the fixed point of the Bellman optimality operator, a stronger criterion that is not well defined for the inverse problem. To alleviate this bias in IRL, we introduce an alternative training loss that puts more weights on future states which yields a reformulation of the (maximum entropy) IRL problem. The algorithms we devised exhibit enhanced performances (and similar tractability) than off-the-shelf ones in multiple OpenAI gym environments.

## 1 INTRODUCTION

Modelling the behaviours of rational agents is a long active research topic. From early attempts to decompose human and animal locomotion Muybridge (1979) to more recent approaches to simulate human movements Li & Todorov (2006); Mombaur (2009); Schultz & Mombaur (2009), the common thread is an underlying assumption that the agents are acting according to some stationary policies. To rationalise these behaviours, it is natural to assume that they are optimal with respect to some objective function (there are evidences to back this assumption in the case of animal conditioned learning Schmajuk & Zanutto (1997); Verschure & Althaus (2003); Maia (2010); Verschure et al. (2014)).

The global objective of Inverse Reinforcement Learning (IRL) is inferring such objective function given measurements of the rational agent's behaviour, its sensory inputs and a model of the environment Russell (1998). IRL builds upon the standard Reinforcement Learning (RL) formulation, where the goal is to find the policy that minimises discounted cumulative costs of some Markov Decision Process Puterman (2014). It aims at finding cost functions for which the observed behaviour is "approximately optimal". However this simplistic formulation admits degenerate solutions Abbeel & Ng (2004). This led to a series of innovative reformulations to lift this indeterminacy by favouring costs for which the observed behaviour is particularly better than alternative ones, namely maximum margin IRL Ratliff et al. (2006) and maximum entropy IRL Ziebart et al. (2008; 2010). The latter formulation ended up as the building block of recent breakthroughs, with both tractable and highly performing algorithms Finn et al. (2016); Ho & Ermon (2016); Fu et al. (2017). These improvements provided the ground for multiple practical real-life applications Ziebart et al. (2008); Bougrain et al. (2012); Sharifzadeh et al. (2016); Jarboui et al. (2019); Martinez-Gil et al. (2020).

We propose an orthogonal improvement to this literature. We question the very pertinence of characterising optimality w.r.t. the cumulative discounted costs as it induces a bias against policies with longer mixing times. We propose an extension of this criterion to alleviate this issue. From this novel objective, we derive reformulations for both the RL and IRL problems. We discuss the ability of existing RL algorithms to solve this new formulation and we generalise existing IRL algorithms to

solve the problem under the new criterion. We back up our proposition with empirical evidence of improved performances in multiple OpenAI gym environments.

## 2    GENERALISED OPTIMALITY CRITERION

In this section, we introduce the classical settings of RL and IRL, as well as the new generalised settings we introduce to alleviate some inherent biases of current methods.

### 2.1    A CLASSICAL RL SETTING

Consider an infinite horizon Markov Decision Process (MDP) $\mathcal{M} = \{\mathcal{S}, \mathcal{A}, \mathcal{P}, c, \gamma, p_0\}$, where:
– $\mathcal{S}$ is either a finite or a compact subset of $\mathbb{R}^d$, for some dimension $d \in \mathbb{N}$

– $\mathcal{A}$ is either a finite or a compact subset of $\mathbb{R}^{d'}$, for $d' \in \mathbb{N}$

– $\mathcal{P}$ is the state transition kernel, i.e., a continuous mapping from $\mathcal{S} \times \mathcal{A}$ to $\Delta(\mathcal{S})$, where $\Delta(\cdot)$ denotes the set of probability measures[1] over some set,

– $c : \mathcal{S} \times \mathcal{A} \to \mathbb{R}$ is a continuous non-negative cost function,

– $p_0 \in \Delta(\mathcal{S})$ is the initial state distribution, and $\gamma \in (0, 1)$ is the discount factor.

A policy $\pi$ is a mapping indicating, at each time step $t \in \mathbb{N}$, the action $a_t$ to be chosen at the current state $s_t$; it could depend on the whole past history of states/actions/rewards but it is well known that one can focus solely, at least under mild assumptions, on stationary policies $\pi : \mathcal{S} \to \Delta(\mathcal{A})$. The choice of a policy $\pi$, along with a kernel $\mathcal{P}$ and the initial probability $p_0$, generates a unique probability distribution over the sequences of states denoted by $\mathbb{P}_\pi$ (the solution to the forward Chapman–Kolmogorov equation). The expected cumulative discounted cost of this policy, in the MDP $\mathcal{M}$ is consequently equal to $\mathbb{E}_{p_0,\pi}[\sum_t \gamma^t c(s_t, a_t)] = \int_{s_0} p_0(s_0) \sum_{t=0}^\infty \int_{s_t,a_t} \gamma^t \mathbb{P}_\pi(s_t, a_t|s_0) c(s_t, a_t)$.

Optimal policies are minimisers of this quantity (existence is ensured under mild assumptions Puterman (2014)). A standard way to compute optimal policies, is to minimise the state-action value mapping defined as: $Q_\pi^c(s, a) = c(s, a) + \sum_{t=1}^\infty \int_{s_t,a_t} \gamma^t \mathbb{P}_\pi(s_t, a_t|s) c(s_t, a_t)$. Indeed, the expected cumulative discounted cost of a policy is the expectation of $Q$-function against $p_0$:

$$\mathbb{E}_{p_0,\pi}[\sum_{t=0}^\infty \gamma^t c(s_t, a_t)] = \int_{s_0,a_0} p_0(s_0)\pi(a_0|s_0)Q_\pi^c(s_0, a_0)$$

### 2.2    A BUILT-IN BIAS IN THE IRL FORMULATION

The problem gets more complicated in Inverse Reinforcement Learning where the objective is to learn an unknown cost function $c$ whose associated optimal policy coincides with a given one $\pi_E$ (referred to as the "expert" policy). This problem is unfortunately ill-posed as all policies are optimal w.r.t. a constant cost function Abbeel & Ng (2004). In order to lift this indeterminacy, the most used alternative formulation is called *maximum entropy inverse reinforcement learning* Ziebart et al. (2008; 2010) that aims at finding a cost function $c^*$ such that the expert policy $\pi_E$ has a relatively small cumulative cost $\mathbb{E}_{p_0,\pi_E}[\sum_{t=0}^\infty \gamma^t c^*]$ while other policies incur a much higher cost. This implicitly boils down to learning an optimal policy (associated to some learned cost) that matches the expert's future state occupancy measure $\rho_{\pi_E}$ marginalised over the initial state distribution, where $\rho_\pi(s, a|s_0) = \sum_t \gamma^t \mathbb{P}_\pi(s_t = s, a_t = a|s_0)$.

State of the art approaches Ho & Ermon (2016); Fu et al. (2017) consist, roughly speaking, in a two-step procedure. In the first step, given a cost function $\hat{c}$, an (approximately) optimal policy $\hat{\pi}$ of $\hat{\mathcal{M}}$ (the MDP $\mathcal{M}$ with $\hat{c}$ for cost function), is learned. In the second step, trajectories generated by $\hat{\pi}$ are *compared* to expert ones (in the sense of $\rho_\pi$); then $\hat{c}$ is updated to penalise states unvisited by the expert (say, by gradient descent over some parameters). Obviously, those two steps can be repeated until convergence (or until the generated and the original data-sets are close enough).

However, the presence of a discount factor in the definition of $\rho_\pi$ has a huge undesirable effect: the total weight of the states in the far future (say, after some stage $t^*$) is negligible in the global

---

[1]The $\sigma$-field is always the Borel one.

loss, as it would be of the order of $\gamma^{t^*}$. So trying to match the future state occupancy measure will implicitly favours policies mimicking the behaviour in the short term. As a consequence, this would end up in penalising policies with longer mixing times even if their stationary distribution matches the experts on the long run. This built-in bias is a consequence of solving the reinforcement learning step with policies that optimise the cumulative discounted costs (minimises the expectation of the Q-functions against $p_0$) rather than policies that achieve the Bellman optimality criterion (minimises the Q-function for any state action pairs). Unfortunately, there is no IRL framework solving the problem under the latter assumption.

In order to bridge this gap, we introduce a more general optimality criterion for the reinforcement learning step; it is still defined as the expectation of the $Q$-function, yet not against $p_0$ as in traditional RL, but against **both** the initial **and** the future states distributions. To get some flexibility, we allow the loss to weight present and future states differently by considering a probability distribution $\eta$ over $\mathbb{N}$. Formally, we define the $\eta$-weighted future state measurement distribution:

$$P_\pi^\eta(s_+, a_+|s_0) := \sum_{n=0}^\infty \eta(n)\mathbb{P}_\pi(s_n = s_+, a_n = a_+|s_0).$$

Using $P_\pi^\eta$, the new criterion is defined as:

$$\mathbb{E}_{p_0,\pi}^\eta[Q_\pi^c] := \int_{s_0} p_0(s_0)\int_{s_+,a_+} P_\pi^\eta(s_+, a_+|s_0)Q_\pi^c(s_+, a_+) = \mathbb{E}_{p_0,\pi}\Big[\sum_k \eta(k)\sum_t \gamma^t c_{t+k}\Big]$$

where $c_t$ denotes the cost at the $t^{\text{th}}$ observation ($c(s_t, a_t)$). Any policy that minimises $\mathbb{E}_{p_0,\pi}^\eta[Q_\pi^c]$ will now be referred to as "$\eta$-optimal" (w.r.t. the cost function $c$). As mentioned before, the inverse RL problem can be decomposed in two sub-problems, learning approximate optimal strategies (given a candidate $\hat{c}$) and optimizing over $\hat{c}$ (taking into account the expert distribution $\pi_E$). In order to avoid over-fitting when learning optimal policies, the standard way is to regularize the optimization loss Geist et al. (2019). As a consequence, we consider any mapping $\Omega : \Delta(\mathcal{A})^{\mathcal{S}} \to \mathbb{R}$ that is a concave over the space of policies. The associated regularised loss of adopting a policy $\pi$ given the cost function $c$ is defined as:

$$\mathcal{L}_\Omega^\eta(\pi, c) = \mathbb{E}_{p_0,\pi}^\eta[Q_\pi^c] - \Omega(\pi) \tag{1}$$

The generalised RL problem is then defined as:

$$\text{RL}_\Omega^\eta(c) := \arg\min_\pi \mathcal{L}_\Omega^\eta(\pi, c) \tag{2}$$

Similarly, in order to learn simpler cost functions Ho & Ermon (2016), the optimization loss considered is in turn penalised by a convex (over the space of cost functions) regularizer $\psi : \mathbb{R}^{(\mathcal{S} \times \mathcal{A})} \to \mathbb{R}$. The problem of Generalised (Maximum Entropy) Inverse Reinforcement learning, whose objective is to learn an appropriate cost function $c$, is formally defined as :

$$\text{IRL}_{\psi,\Omega}^\eta(\pi_E) := \arg\max_c \min_\pi \mathcal{L}_\Omega^\eta(\pi, c) - \mathcal{L}_\Omega^\eta(\pi_E, c) - \psi(c) \tag{3}$$

We emphasise that simply choosing $\delta_0$ (a Dirac mass at 0) for the distribution $\eta$ induces the classical definitions of both the RL and IRL problems Ho & Ermon (2016). On the other hand, choosing $\eta = \text{Geom}(\gamma)$ transforms the loss into the expectation of the sum of discounted Q-functions along the trajectory.

Hypothetically, there could be other generalisations of discounted cost. However, preserving the compatibility of the Bellman criterion with the proposed generalisation for RL and duality properties for IRL is not trivial (for example, polynomial decay $\frac{\gamma}{t^n}$ would break these properties). In the following, we prove that the $\eta$-optimality framework satisfies both properties.

## 2.3 GENERALISED REINFORCEMENT LEARNING

As in the classical setting, solving the generalised IRL problem (Equation 3), requires solving the generalised RL problem (Equation 2) as a sub-routine. Among the model free RL algorithms, value-based vs. policy gradient-based methods can be distinguished. In this section, we focus on the first type of methods as they can easily be used for the search of $\eta$-optimal policies. We provide a detailed discussion of the limitations of current policy gradient-based methods in Appendix A, as they might be less adapted to solving $\text{RL}_\Omega^\eta$.

Given a standard MDP $\mathcal{M}$ and policy $\pi$, the Bellman operator $T_\pi$ (from $\mathbb{R}^\mathcal{S}$ to $\mathbb{R}^\mathcal{S}$) is defined as Geist et al. (2019):

$$[T_\pi(v)](s) = \mathbb{E}_{a \sim \pi}\Big[c(s,a) + \gamma \mathbb{E}_{s'|s,a}[v(s')]\Big],$$

and its unique fixed point is called the associated value function $v_\pi^c$. This concept is transposed to the regularised case as follows: Given a concave regularisation function $\Omega$ and a policy $\pi$, the associated regularised Bellman operator $T_{\pi,\Omega}$ and the associated value function $v_{\pi,\Omega}^c$ are respectively defined as:

$$T_{\pi,\Omega} : v \in \mathbb{R}^\mathcal{S} \rightarrow T_{\pi,\Omega}(v) = T_\pi(v) - \Omega(\pi) \in \mathbb{R}^\mathcal{S},$$

and as: $v_{\pi,\Omega}^c = T_{\pi,\Omega}(v_{\pi,\Omega}^c)$, its unique fixed point. As usual, the regularised Bellman *optimality* operator $T_{*,\Omega}$ is in turn defined as:

$$T_{*,\Omega} : v \in \mathbb{R}^\mathcal{S} \mapsto T_{*,\Omega}(v) \in \mathbb{R}^\mathcal{S}$$
$$[T_{*,\Omega}(v)](s) = \min_\pi [T_{\pi,\Omega}(v)](s), \quad \forall s \in \mathcal{S}.$$

Notice that given $v \in \mathbb{R}^\mathcal{S}$, the policy $\bar{\pi}_v(\cdot|s) = \delta_{\bar{a}}$ with $\bar{a} = \arg\min_a c(s,a) + \gamma \mathbb{E}_{s'|s,a}[v(s')]$ achieves the minimum in the overall equation for all state $s \in \mathcal{S}$. The policy improvement theorem Sutton & Barto (2018) guarantees that if $v_{\pi,\Omega}^c$ is the regularised value function of $\pi$, then $\bar{\pi} := \bar{\pi}_{v_{\pi,\Omega}^c}$ dominates $\pi$ (in the sense that $v_{\bar{\pi},\Omega}^c(s) \leq v_{\pi,\Omega}^c(s)$ for any state $s \in \mathcal{S}$).

If we denote by $v_{*,\Omega}^c$ the unique fixed point of the regularised Bellman optimality operator $T_{*,\Omega}$, then the policy $\pi_\Omega^* := \bar{\pi}_{v_{*,\Omega}^c}$ is associated to the minimum regularised value function:

**Proposition 1** *Optimal regularised policy (Theorem 1 of Geist et al. (2019)) : The policy $\pi_\Omega^* := \bar{\pi}_{v_{*,\Omega}^c}$ is the unique optimal regularised policy in the sense that, for all policies $\pi$, the following holds:*

$$\forall s \in \mathcal{S}, \quad v_{\pi_\Omega^*,\Omega}^c(s) = v_{*,\Omega}^c(s) \leq v_{\pi,\Omega}^c(s).$$

Notice how the optimal regularised policy $\pi_\Omega^*$ (that minimizes the regularised cumulative discounted cost), is intuitively a good proxy of optimal policies in the sense of the regularised $\eta$-weighted Q-functions (as it optimises the Q-function for all state action pairs and therefor indirectly optimises the $\eta$-weighted Q-functions).

For this reason, we propose to exploit state of the art value base RL algorithms that optimise $v_{*,\Omega}$ (such as Soft Actor Critic (SAC) Haarnoja et al. (2018)) to approximately solve the generalised setting. Naturally, the optimality gap depends directly on the exact form of the distribution $\eta$. Unfortunately, there is no known performance bound for such approximation. However, we found out empirically that for Geometric and Poisson $\eta$ distributions the derived IRL algorithms for the generalised setting (which we aim to solve in this paper) did not suffer from this approximation.

## 2.4 GENERALISED INVERSE REINFORCEMENT LEARNING

We recall that the global objective of IRL is to learn the cost function based on an expert policy $\pi_E$. In this section, we illustrate that the solution of $\mathrm{IRL}_{\psi,\Omega}^\eta(\pi_E)$ is a cost function $\hat{c}$, whose associated optimal policy $\mathrm{RL}_\Omega^\eta(\hat{c})$ matches the expert's future state distributions $P_{\pi_E}^\eta$ marginalised against $\rho_{\pi_E}$ rather than simply matching the occupancy measure $\rho_{\pi_E}$, as in usual IRL formulation. To alleviate notations, we denote the $\eta$-optimal policy $\hat{\pi} = \mathrm{RL}_\Omega^\eta(\hat{c})$ as $\mathrm{RL}_\Omega^\eta \circ \mathrm{IRL}_{\psi,\Omega}^\eta(\pi_E)$. This policy minimises the worst-case cost weighted divergence $d_c(\hat{\pi}\|\pi_E) : \mathcal{S} \mapsto \mathbb{R}$ averaged over $p_0$, such that:

$$d_c(\hat{\pi}\|\pi_E)(s_0) := \int_{s,a,s_+,a_+} c(s_+,a_+)\Big[\rho_{\hat{\pi}}(s,a|s_0)P_{\hat{\pi}}^\eta(s_+,a_+|s,a) - \rho_{\pi_E}(s,a|s_0)P_{\pi_E}^\eta(s_+,a_+|s,a)\Big]$$

This is formalised in the following proposition that requires the following notations. Given a policy $\pi$ we denote by $\mu_\pi(s_+,a_+|s_0) = \sum_{t,k}\gamma^t\eta(k)\mathbb{P}_\pi(s_{t+k} = s_+, a_{t+k} = a_+|s_0)$ the frequency of $(s_+,a_+)$ in the $\eta$-weighted future steps of trajectories initialised according to $\rho_\pi(s,a|s_0)$.

**Proposition 2** *For any convex penalty $\psi$, concave regulariser $\Omega$ (w.r.t. the future occupancy measure $\mu_\pi$) and any expert policy $\pi_E$, if $\eta$ is geometric, then:*

$$\mathrm{RL}_\Omega^\eta \circ \mathrm{IRL}_{\psi,\Omega}^\eta(\pi_E) = \arg\min_\pi \max_c L(\pi,c)$$

$$\text{where: } L(\pi,c) = -\Omega(\pi) - \psi(c) + \int_{s_0} p_0(s_0)d_c(\pi\|\pi_E)(s_0)$$

As mentioned before, Proposition 2 states that in its generalised formulation, solving the IRL problem can be done by matching $\eta$-weighted future state distributions $\mu_\pi$ (as opposed to matching $\rho_\pi$ in the classical case). This proves that the generalised setting preserves the duality properties of classical IRL. The solution of $\mathrm{IRL}_{\psi,\Omega}^\eta$ is a Nash-Equilibrium of a game between poicies and the cost functions:

**Corollary 2.1** *Under the assumptions of Proposition 2, $(\tilde{c}, \tilde{\pi}) = (\mathrm{IRL}_{\psi,\Omega}^\eta(\pi_E), \mathrm{RL}_\Omega^\eta(\tilde{c}))$ is a Nash-Equilibrium of the following game:*

$$\tilde{c} : \max_c L(\pi, c) + \Omega(\pi) \quad ; \quad \tilde{\pi} : \min_\pi \mathbb{E}_+^\pi \big[ Q_\pi(c) \big] - \Omega(\pi)$$

A practical implication of Corollary 2.1 is a template algorithm dubbed $\mathrm{GIRL}_{(\psi,\Omega,\eta)}$ and illustrated in Algorithm 1 that can be used to solve this problem approximately.

---

**Algorithm 1** $\mathrm{GIRL}_{(\psi,\Omega,\eta)}$ (Generalised IRL)

---

1: **Input:** Expert trajectories $\tau_E \sim \pi_E$, initial policy $\pi_{\theta_0}$ and initial cost function $c_{w_0}$
2: **for** $e \in [1, N]$ **do**
3:      Sample trajectories $\tau \sim \pi_{\theta_i}$
4:      Sample from $\tau$ policy state action $(S^+, A^+) \sim \mu_\pi^\eta(\tau)$
5:      Sample from $\tau_E$ expert state action $(S_E^+, A_E^+) \sim \mu_{\pi_E}^\eta(\tau_E)$
6:      Update the cost parameter $w_i$ to maximise $-\psi(c_w) + \sum_{S^+, A^+} c_w(s, a) - \sum_{S_E^+, A_E^+} c_w(s, a)$
7:      Update $\theta_i$ using a value-based reinforcement learning algorithm to minimise $c_{w_{i+1}}$
8: **Return:** $(\pi_{\theta_N}, c_{w_N})$

---

We stress out now that the concavity of $\Omega$ w.r.t. $\mu_\pi$ in Proposition 2 is not too restrictive in practical settings as the $\eta$-weighted entropy regulariser, amongst others, satisfies it:

**Proposition 3** *The $\eta$-weighted entropy regulariser $\bar{H}_{p_0}^\eta$ defined by*

$$\bar{H}_{p_0}^\eta(\mu_\pi) := H_{p_0}^\eta(\pi) = \mathbb{E}_{p_0,\pi}^\eta \Big[ \sum_t -\gamma^t \log \big[ \pi(a_t|s_t) \big] \Big]$$

*is concave with respect to the occupancy measure $\mu_\pi$.*

## 3 TRACTABILITY

The tractability of $\mathrm{GIRL}_{\psi,\Omega,\eta}$ is a crucial requirement for practical implementation. In this section, both the regulariser term $\Omega(\cdot)$ and the penalty term $\psi(\cdot)$ are assumed to be tractably optimisable. For example the entropy, a widely used regulariser in the RL literature is efficiently tractable in practice. Indeed, Soft Actor Critic Haarnoja et al. (2018) uses a single sample approximation of the entropy to optimise the entropy regularised Bellman optimality operator. Similarly, using an indicator penalty over a subset $\mathcal{C}$ of possible cost functions (i.e., the penalty is infinite if $c \notin \mathcal{C}$ and 0 otherwise) is also tractable with projected gradient updates if $\mathcal{C}$ is convex Abbeel & Ng (2004); Syed et al. (2008); Syed & Schapire (2007). As a consequence, establishing tractability of $\mathrm{GIRL}_{\psi,\Omega,\eta}$ reduces to finding tractable sampling schemes from $\mu_\pi$. This is equivalent to sampling sequentially from the $\eta$-weighted future state distribution and the occupancy measure as:

$$\mu_\pi(s_+, a_+|s_0) = \int_{s,a} \rho_\pi(s, a|s_0) P_\pi^\eta(s_+, a_+|s, a) = \int_{s,a} P_\pi^\eta(s, a|s_0) \rho_\pi(s_+, a_+|s, a)$$

Given a policy $\pi$, the simplest approach to sample from these distributions is to sample transitions from a set of $\pi$-generated trajectories, denoted by $\{(s_t^{(i)}, a_t^{(i)})_{t \in \{1, H\}}; i \in \{1, N\}\}$, where $H$ is the horizon and $N$ is the number of trajectories.

**– For the occupancy measure $\rho_\pi$:** given a uniformly sampled index $i \sim \mathcal{U}[1, N]$ and a time sampled from truncated geometric distribution $t \sim \mathrm{Geom}_{[1,H]}(\gamma)$, the associated pair of state/action $(s_t^{(i)}, a_t^{(i)})$ is an (approximate) sample from the marginal of $\rho_\pi(.|s_0)$ against $p_0$.
**– For the future state distribution $P_\pi^\eta$:** Given a state $s_t^{(i)}$ sampled as above, a time $k$ is sampled from a truncated $\eta_{[1,H-t]}$; the state-action $(s_{t+k}^{(i)}, a_{t+k}^{(i)})$ is an approximate sample from $P_\pi^\eta(.|s_t)$.

As a consequence, the above scheme shows that sampling from $\rho_\pi$ and $P_\pi^\eta$ reduces to sampling indices from $\text{Geom}(\gamma)$ and $\eta$, which is tractable from both the expert and the learned policies perspective. This proves that solving $\text{IRL}_{\psi,\Omega}^\eta$ does not incur any additional computational burden.

## 4    MEGAN: MAXIMUM ENTROPY - GENERATIVE ADVERSARIAL NETWORK

This section introduces a new algorithm, called MEGAN, that will improve upon state of the art IRL algorithms. Recent progresses in the field propose variations of GAIL Ho & Ermon (2016) in order to solve a wide variety of problems. For example, AIRL Fu et al. (2017) uses a particular shape for the discriminator for better transferability of the learned rewards, EAIRL Qureshi et al. (2018) applies empowerment regulariser to policy updates to prevent over-fitting the expert demonstration, RAIRL Jeon et al. (2020) generalises AIRL for regularised MDPs (i.e. $\Omega$ is not necessarily the entropy), s-GAIL Kobayashi et al. (2019) generalises the formulation for multi-task RL, etc.

Their contributions were crucial to the progresses of IRL. However, we will actually focus on improving the core algorithm GAIL so that all the aforementioned approaches can be implemented with MEGAN instead of GAIL with improved performances.

We considered the rather classical penalty function Ho & Ermon (2016):

$$\psi_{GAN}(c) = \left\{ \begin{array}{ll} \mathbb{E}_{p_0,\pi_E}^\eta[g(c(s,a))] & \text{if } c < 0 \\ +\infty & \text{otherwise} \end{array} \right. \quad \textbf{where: } g(x) = \left\{ \begin{array}{ll} -x - \log(1 - e^x) & \text{if } x < 0 \\ +\infty & \text{if } x \geq 0 \end{array} \right.$$

The generalised problem boils down to using $\mathbb{E}_\pi^\eta$ instead of $\mathbb{E}_\pi$:

**Proposition 4** *Under the assumptions of Proposition 2, and for $\psi = \psi_{GAN}(c)$, it holds:*

$$\text{RL}_\Omega^\eta \circ \text{IRL}_\psi^\eta(\pi_E) = \arg\min_\pi -\Omega(\pi) + \max_{D \in (0,1)^{S \times A}} \mathbb{E}_\pi^\eta[\log D] - \mathbb{E}_{\pi_E}^\eta[\log(1 - D))]$$

The algorithm MEGAN (Maximum Entropy - Generative Adversarial Network), is then equivalent to $\text{GIRL}_{\psi_{GAN},H,\text{Geom}(\gamma)}$ and a generalisation of the corner stone in state of the art IRL Ho & Ermon (2016); its pseudo-code is given in Algorithm 2.

---

**Algorithm 2** MEGAN

1: **Input:** Expert trajectories $\tau_E \sim \pi_E$, initial policy $\pi_{\theta_0}$ and initial discriminator function $D_{w_0}$
2: **for** $e \in [1, N]$ **do**
3:     Sample trajectories $\tau \sim \pi_{\theta_i}$
4:     Sample states randomly $(S_t, A_t) \sim \tau$ and $(S^+, A^+) = (S_{t+k}, A_{t+k})$ where $k \sim \eta$
5:     Sample states randomly $(S'_t, A'_t) \sim \tau_E$ and $(S_E^+, A_E^+) = (S'_{t+k}, A'_{t+k})$ where $k \sim \eta$
6:     Update the cost parameter $w_i$ to maximise $\left[ \log D_w(S^+, A^+) - \log(1 - D_w(S_E^+, A_E^+)) \right]$
7:     Update $\theta_i$ using soft actor critic to minimise the cost $\left[ \log D_{w_{i+1}} \right]$
8: **Return:** $(\pi_{\theta_N}, D_{w_N})$

---

## 5    EXPERIMENTS

This section is devoted to experimental evidences that MEGAN achieves state of the art performances. It is compared to GAIL as all subsequent approaches build upon its formulation. The standard approach to compare IRL algorithms is to consider the best performing policies obtained during the training and evaluate their performances. This is an issue in practice as we do not have access to such cost function in order to implement a stopping rule once the learned policy reaches a certain performance threshold. A reasonable alternative criterion is to measure the divergence between generated and expert future state distributions (in the sense of $\rho_\pi$ or $\mu_\pi$). In this section, we propose to evaluate the divergence using the maximum mean discrepancy (MMD)[2]. We will tackle the following questions empirically:

---

[2]A formal reminder on the definition of MMD divergence is provided in Appendix E.1 for completeness.

**-1** How does varying the parameter of a geometric $\eta$ distribution affect performances?

**-2** How does alternative $\eta$ distribution (e.g. a Poisson) compare to the use of a geometric one?

**-3** Does varying the discount factor $\gamma$ produce similar performances?

Due to limited space, we only analyse single-task environments in this section. We provide in Appendix C and D further investigations for the multi-task setting. A summary of the used hyperparameters is also provided in Appendix E.2.

## 5.1 PERFORMANCE IMPROVEMENT USING A GEOMETRIC $\eta$ DISTRIBUTION

Recall that solving the IRL problem essentially boils down to finding an equilibrium between a policy that matches the expert behaviour and a cost function that discriminates generated trajectories from expert ones. An important property of a given algorithm is the stability of the associated equilibrium. In order to take into account this aspect, we propose to evaluate performances using trajectories sampled during the last 100 iterations of training. We will refer to these trajectories as the remaining replay buffer. This procedure provides an evaluation of the policies toward which the algorithm converges, while factoring in their stability.

Notice that the goal of GAIL is to match the distribution $\rho_{\pi_E}$ while MEGAN matches the distribution $\mu^{\eta}_{\pi_E}$. In order to take into account this difference, we propose to measure performances in terms of cumulative costs, $\mathrm{MMD}_{\rho} = \mathrm{MMD}(\rho_{\pi}, \rho_{\pi_E})$ and $\mathrm{MMD}_{\mu} = \mathrm{MMD}(\mu^{GEOM(0.99)}_{\pi}, \mu^{GEOM(0.99)}_{\pi_E})$.

We evaluate the performances of MEGAN using a truncated[3] geometric $\eta$ distribution with different parameters (specifically $\{0, 0.25, 0.5, 0.75, 1\}$). Notice that using a geometric distribution with parameter 0 is equivalent to using a Dirac mass at 0 (or equivalently solving the IRL problem using GAIL). Similarly, using a geometric distribution with parameter 1 is equivalent to using a uniform $\eta$ distribution. The remaining values can be seen as an interpolation between these extremes.

In Figure 1, each point reports the average performances obtained using the remaining replay buffer of 3 randomly seeded instances of the algorithm. The blue curves report the average $\mathrm{MMD}_{\rho}$ (divergence in the sense of the classical IRL formulation), the green curves report the average $\mathrm{MMD}_{\mu}$ (divergence in the sense of the generalised IRL formulation), and the red curves report the average cumulative costs (divergence in the sense of the environment's ground truth). From left to right, we report the performances in three classical control settings with varying complexity from the *MuJoCo* based environments Plappert et al. (2018). In Figure 1a we used the Ant environment (a state action space of dimension 118), in Figure 1b we used the Half-Cheetah environment (a state action space of dimension 23) and in Figure 1c we used the Hopper environment (a state action space of dimension 14).

All the provided experiments confirmed a reduction of the average MMD divergence by 25% to 60% (in the sense of both classical IRL and generalised IRL formulations) as the parameter of the $\eta$ distribution increased to 1. This confirms that using the $\eta$-optimality objective function improves both the stability and the ability of IRL algorithms to match faithfully the expert behaviour. Notice that despite the fact that GAIL explicitly optimises divergence in the sense of $\rho_{\pi}$, it under-performs in the sense of $\mathrm{MMD}_{\rho}$ when compared to MEGAN (the blue curves decreases as the parameter of the geometric distribution increases). This confirms empirically that the $\eta$-optimality framework proposed in this paper does indeed bridge the gap between policy-based reinforcement learning (optimising cumulative discounted costs) and value-based reinforcement learning (achieving the Bellman optimality criterion) as it even improves performances in the sense of classical IRL.

Another important observation in Figure 1, is that for complex environments (Ant and Half-Cheetah) the decrease of the MMD divergence -as we increased the parameter of the geometric distribution to 1- was correlated with a decrease of the average cumulative costs by a factor of 2 to 4. This was not the case of the Hopper environment, as we obtained similar cumulative costs despite the reduction of the divergence by a factor of 3. This is explained by the fact that the ground truth cost function of the Hopper environment produces similar cumulative costs for a wider variety of policies. For this reason, the IRL solution does not need to achieve a faithful expert behaviour matching in order to achieve good performances. This illustrates the importance of evaluating IRL algorithm with respect to $\mathrm{MMD}_{\rho}$ and $\mathrm{MMD}_{\mu}$ when the goal is to mimic behaviors.

---

[3]Due to obvious computational limitations, the trajectories are finite

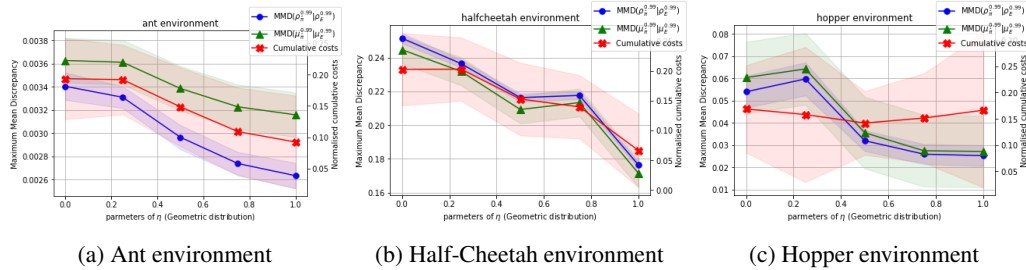

(a) Ant environment       (b) Half-Cheetah environment       (c) Hopper environment

Figure 1: **Performances of the policies obtained during the last** 100 **iteration of** MEGAN**:** as the parameter of $\eta = \mathrm{Geom}(\kappa)$ grows from 0 (equivalent to an instance of GAIL) to 1 (equivalent to an instance of MEGAN with a uniform $\eta$), we observe that the learned policies' generated trajectories are increasingly similar with those generated by the expert in the sense of $\rho_\pi$ (classical IRL criterion), $\mu_\pi$ (Generalised IRL criterion) and the cumulative discounted costs (The environment's ground truth).

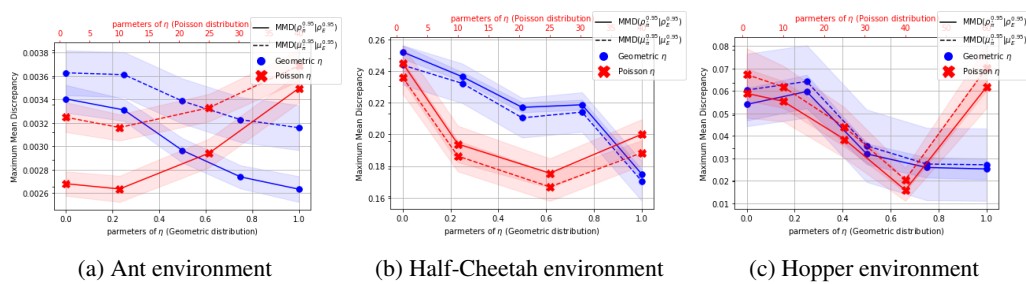

(a) Ant environment       (b) Half-Cheetah environment       (c) Hopper environment

Figure 2: **Performances of** MEGAN **using a Poisson** $\eta$ **distribution:** Setting the parameter of $\eta = \mathrm{Poisson}(\lambda)$ to a value around the length of the expert's movement cycle achieved similar/better performances than those obtained using a uniform $\eta$ distribution ($\mathrm{Geom}(1)$). The expert cycle is roughly 10 frames long in the Ant environment, 25 in the Half-Cheetah, and 40 in the Hopper.

## 5.2 Performance improvement using a Poisson $\eta$ distribution

Using the same experimental setting from the previous section, we evaluate MEGAN's performances with non-geometric $\eta$ distributions. In Figure 2, we plot the performances of the remaining replay buffer obtained with geometric $\eta$ distributions in blue lines, and those obtained when using a Poisson distribution in red. To reduce clutter, we removed the cumulative costs and only provided the divergences $\mathrm{MMD}_\rho$ (represented with solid lines in Figure 2) and $\mathrm{MMD}_\mu$ (dashed lines in Figure 2).

Despite the weaker theoretical guarantees provided in our paper for the case of non-geometric $\eta$ distributions, we observe that using a Poisson $\eta$ can lead to comparable performances. Recall that the expectation of a Poisson distribution is equal to its parameter value. This implies that solving $\mathrm{IRL}_{\psi,\Omega}^{Poisson(\lambda)}$ searches for policies that match $\rho_{\pi_E}(.|s)$ for states $s$ observed around the $\lambda^{th}$ frame of the expert demonstrations. Now notice that the control tasks analysed in Figure 2, consist of movements cycles that are repeated perpetually. Quite interestingly, setting $\lambda$ to a value around the length of an expert cycle ($\lambda = 10$ in the Ant environment, 25 in the Half-Cheetah, and 40 for the Hopper), ended up achieving the best performances.

In a sense, the proposed $\eta$-optimality criterion can be seen as an inductive bias: we successfully injected qualitative knowledge (the repetitive nature of the expert behaviour) by explicitly asking the agent to focus on matching $\rho_\pi(.|s)$ for states $s$ observed within a single movement cycle of the expert demonstrations via careful parameterisation of the distribution $\eta$. In the case where such higher understanding/representation of the expert behaviour is unavailable, using a uniform $\eta$ distribution (or a geometric $\eta$ with a parameter close to 1) is a safe bet. Notice that in Figure 2, the both the blue and red curves have comparable minimum values.

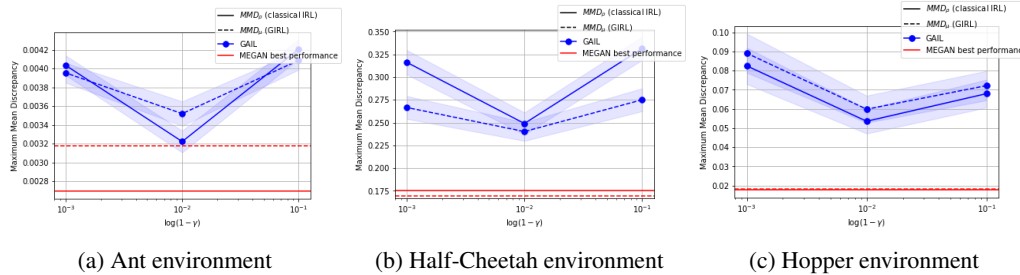

(a) Ant environment      (b) Half-Cheetah environment      (c) Hopper environment

Figure 3: **Performances of** GAIL **as we vary the discount factor** $\gamma$**:** Neither increasing nor decreasing the discount factor resulted in improved performances. Agnostic of the used parameter, GAIL was not able to match the expert behavior as well as MEGAN.

### 5.3 EFFECT OF VARYING THE DISCOUNT FACTOR $\gamma$

Recall that in the classical problem formulation, the discount factor $\gamma$ can be interpreted as the weight of future observations. In this section, we investigate whether changing this parameter can overcome the buit-in bias against policies with longer mixing times, without resorting to the $\eta$-optimality criterion. In practice, the discount factor $\gamma$ is used separately in two building blocks of IRL algorithms (including GAIL and MEGAN). The first instance is in the Bellman updates when solving the RL problem under a given cost function: we refer to this parameter as $\gamma_{\mathrm{RL}}$. The second instance is in the discrimination problem when approximately sampling from $\rho_\pi$: we refer to this parameter as $\gamma_{\mathrm{IRL}}$.

Due to the finite nature of the expert demonstrations, the standard approach is to approximate future state distributions by setting $\gamma_{\mathrm{IRL}}$ to 1 (or equivalently, sampling transitions uniformly). This can be seen as an asymptotic behaviour of the classical problem formulation: as the discount factor approaches 1, the associated truncated geometric distribution will converge to a uniform one. Reducing this parameter will only accentuate the discussed issues as it entails up-sampling the early stages of the collected demonstrations, which will inevitably favor short term imitation. On the other hand, it is not possible to set $\gamma_{\mathrm{RL}} = 1$ (the Bellman operator is no longer guaranteed to admit a fixed point, and state of the art value based RL algorithms become extremely unstable). For this reason, most practitioners set this parameter to a value reasonably close to 1.

In this section, we evaluated the remaining replay buffers obtained using GAIL as we vary the discount factors values ($\gamma_{RL} \in [0.9, 0.99, 0.999]$). We emphasize that in all the reported empirical evaluations (including previous ones, i.e. Figures 1 and 2), we fixed $\gamma_{\mathrm{IRL}}$ to 1.

In Figure 3, we report the average $\mathrm{MMD}_\rho$ divergence of the remaining replay buffer in solid lines, and the average $\mathrm{MMD}_\mu$ divergence in dashed lines. GAIL's performances as we vary the discount factor are reported in blue and the best performances obtained with MEGAN are reported in red. Reducing $\gamma_{RL}$ to 0.9 accentuated the bias against policies with longer mixing times, and on the other hand increasing it to 0.999 lead to a less reliable RL algorithm. As expected, we observe in Figure 3 that both tweaks did not entail performances on par to what we obtained using MEGAN.

## 6 CONCLUSION

In this paper, we generalised the classical criterion of optimality in the reinforcement learning literature by putting more weights onto future observations. Using this novel criterion, we reformulated both the regularised RL and the maximum-entropy IRL problems. We reviewed existing RL algorithms and discussed their ability to search for $\eta$-optimal policies. We also generalised classical IRL solutions. The derived algorithm produced stable solutions with enhanced expert matching properties.

In practice, the main difference between MEGAN and GAIL is the discriminator's sampling procedure. This implies that it can easily replace the latter algorithm in all subsequent contributions. An interesting future direction of research consists in evaluating the margin of improvement that can be gained from this modification.

BROADER IMPACT STATEMENT

Inverse reinforcement learning provides a framework to explain observed complex behaviors in sophisticated environments. However, the inherent biases of current formulations againt policies with longer mixing times due to the use of geometrically discounted optimality criterion, prevent existing algorithms from reaching the frameworks full potential. By constructing approximate solutions of the IRL problem (in the sense of the $\eta$-optimality criterion), we believe that we take a step towards more applicable IRL algorithms. In that spirit, we also commit ourselves to releasing our code soon in order to allow the wider community to extend our work in the future.

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
