# OpenReview forum: "A Generalised Inverse Reinforcement Learning Framework"
_ICLR.cc/2022/Conference — ICLR 2022 Submitted_

### Official Review · Reviewer_j2NT · 2021-11-02

**Correctness:** 3
**Technical Novelty And Significance:** 3
**Empirical Novelty And Significance:** 3
**Recommendation:** 6
**Confidence:** 3

**Main Review:**

The issue raised in this paper is novel as far as I know. The proposed solution is interesting and seems to lead to good experimental performance. However, the explanation for why penalizing penalizing longer mixing times is an issue is not clear to me, since two policies regardless of their mixing times that match the stationary distribution of the expert demonstrations should have the same cumulative costs. Does this issue come up because of finite-length trajectories used in RL/IRL? If this explanation was true, shouldn't the experiments with varying discount factors suggest that higher discount factors lead to better results?

The paper should be checked for typos. For instance:

Citations should be in parenthesese, e.g.:
Name et al., (year) -> (Name et al., year)

In Section 2.1, there are some issues with the notations:
In Q^c_\pi, the expectation with respect to P(s_1 | s, a) is missing.

Is matching state occupancy really related to maximum entropy IRL?

Page 3, line 1-2: will implicitly favours -> will implicitly favour

Page 4: therefor

**Summary Of The Paper:**

The paper extends the maximum entropy inverse reinforcement learning (IRL) framework by changing the optimal criterion used in reinforcement learning (RL). This novel criterion is an expectation of the Q-values over a weighted distribution over states and actions induced by a policy, which is in contrast to the standard criterion that is an expectation over the initial state dsitribution.

**Summary Of The Review:**

Novel issue in IRL with interesting solution to tackle it, although the proposition sounds a bit heuristics. A better explanation for it would have made the paper stronger.

---

### Official Review · Reviewer_Msxm · 2021-11-03

**Correctness:** 3
**Technical Novelty And Significance:** 2
**Empirical Novelty And Significance:** 2
**Recommendation:** 5
**Confidence:** 3

**Main Review:**

Overall, I found the premise of the paper intriguing. Although I am not fully up to date with the current IRL literature, the problem addressed in the paper strikes me as novel, and the proposed method (whose derivations I did not check in detail) follows naturally from the formulation proposed. The results also seem promising, although I am not sure that I was able to read them properly (more on this below).

This said, there are several aspects in which I believe the paper could be improved.

- First and foremost is, perhaps, the motivation for the proposed setting. Although the paper makes an argument for the proposed setting based on the bias introduced by the standard optimality criterion underlying IRL, I do not fully grasp how relevant the impact of such bias truly is. In other words, although I understand the technical argument, I am not sure that this translates into an actual problem in practice, and some more discussion on this would be welcome.

- Following on the previous item, and if I understood correctly, it seems to me that the introduced weighting is tantamount to considering a policy-dependent initial distribution $p_0$ in the standard formulation. The different ways of weighting the future state-action pairs correspond to different dependencies on the policy, and perhaps it may be easier to make a case for the proposed approach from that interpretation (just an idea, I don't know if that is the case or not).

- The notation is also a bit overloaded and not always in a very intuitive way. This makes parsing the technical derivations sometimes quite difficult. For example, the symbol "P" appears in several different shapes and sizes with different meanings. As another example, the value function associated with the optimal policy computed from a regularized RL objective is denoted as $v^c_{\pi^*_\Omega,\Omega}$, with all the sub- and super-scripts. Also, I understand that treating the state and action spaces as continuous is convenient for the sake of generality, but the notation used in the resulting integrals is often sloppy and does not add much to the understanding of the paper.

* Finally, the experimental results are somewhat hard to read. Although the plots suggest that, indeed, the performance seems to improve as $\eta$ increases, the differences (as measured in the $yy$ axis of the plots) seem to be quite small. Additionally, except for the half-cheetah environment, the difference between the MMD on $\mu_\pi$ for $\eta=0$ and $\eta=1$ does not seem to be statistically significant. Some more discussion on this aspect would be welcome.

*Minor aspects:*

- Throughout the paper the references do not seem to be properly typeset, which at points is somewhat disruptive.

- Maybe I missed something, but the definition of $\mu_\pi(s_+a_+\mid s_0)$ in the beginning of Section 2.4 is not clear: the paper defines it as the frequency of $(s_+,a_+)$ in the $\eta$-weighted future steps of trajectories initialized according to $\rho_\pi(s,a\mid s_0)$, but I do not see where $\rho_\pi(s,a\mid s_0)$ come into play in the definition.

**Summary Of The Paper:**

The paper proposes a new formulation for inverse reinforcement learning that aims to address the _bias against policies with longer mixing times_. The key contribution of the paper is the proposal of an alternative optimality criterion that arguably addresses the aforementioned bias by considering, for a given policy, the value

$L(\pi,c)=E_{p_0,\pi}\left[\sum_{k=0}^\infty\eta(k)\sum_{t=0}^\infty\gamma^tc_{t+k}\right]$

instead of the more standard definition

$L(\pi,c)=E_{p_0,\pi}\left[\sum_{t=0}^\infty\gamma^tc_{t}\right],$

where $c$ is such that $E[c_t\mid s_t,a_t]=c(s_t,a_t)$.

To solve the IRL problem associated with this new optimality criterion (or, rather, a regularized version thereof) the paper proposes the use of maximum causal entropy IRL, which can roughly be broken down in two subproblems:

- Given a candidate cost function, $\hat{c}$, find a candidate policy, $\hat{\pi}$, that minimizes $L(\hat{\pi},\hat{c})$. In a sense, $\hat{\pi}$ is the "optimal" policy given the cost function $\hat{c}$. The paper proposes the use of soft actor-critic approach (although other value-based approaches could be used).

- Given the expert policy, $\pi_E$, and the candidate policy, $\hat{\pi}$, come up with a candidate cost function $\hat{c}$ such that the expert policy has a low cost and other policies have a high cost, which the paper shows that can be done by minimizing a measure of divergence between the (weighted) distributions induced by $\hat{\pi}$ and $\pi_E$.

The resulting algorithm is a variation of the generative adversarial IRL approach of Ho and Ermon, that the experiments suggest may be able to better recover the expert's policy.

**Summary Of The Review:**

Although the premise of the paper is intriguing, it is not clear to me the relevance of the proposed approach, since the motivation is not convincing.

---

### Official Review · Reviewer_hoAL · 2021-11-04

**Correctness:** 3
**Technical Novelty And Significance:** 2
**Empirical Novelty And Significance:** 2
**Recommendation:** 5
**Confidence:** 4

**Main Review:**

-Originality: GIRL provides a new and interesting perspective for extending the GAIL framework to flexibly balance the importance of short-term and longer-term behavior. I agree that the vanilla GAIL could largely ignore the benefits of longer-term imitation and the distribution $\eta$ could provide a tractable way to address this. One aspect that remains unclear to me is how to properly choose $\eta$ to find the sweet spot of balancing short-term and long-term imitation. The empirical results show that geometric and Poisson distributions work quite well in three MuJoCo tasks, but it remains unknown whether these two types of distributions are universal in improving the matching of occupancy measures. A more rigorous discussion on the choice of $\eta$ is needed.

-Technicity: Though I can appreciate this new perspective, the overall technical novelty is somewhat limited as GIRL mostly follows the algorithmic framework of GAIL.

-Significance: The empirical results show that the proposed GIRL framework does indeed match the state-action occupancy measure of the experts better than GAIL by a large margin.
While the empirical results look very promising, there are several claims that require further explanation:
- As mentioned in Section 5.1, despite that GAIL explicitly optimizes divergence in the standard occupancy measure, GAIL still underperforms MEGAN in this metric. This is indeed quite counterintuitive and would require further explanation.
- Section 5.3 shows that even setting the discount factor $\gamma$ of the RL problem close to 1 could not address long-term imitation. What is the fundamental reason behind this phenomenon (in addition to simply stating “increasing it to 0.999 leads to a less reliable RL algorithm”)?

A more in-depth explanation of the experimental results would be very much appreciated.

-Clarity: The paper is well-written and well-organized. The proposed method is in most places well-explained.

-Other comments:
- There are many candidate metrics for measuring the difference between occupancy measures. Is there any specific reason why MMD is selected?
- The integral in the definition of $E_{p_0,\pi}^{\eta}[Q^{c}_{\pi}]$ is missing the terms $ds_0$, $ds+$, and $da+$?


**Summary Of The Paper:**

This paper points out that the classical IRL approach has a tendency to match those occupancy measures that favor short-term behavior. To address this issue, a reformulation is proposed based on GAIL in order to put more emphasis on matching longer-term behavior. Specifically, the main difference is to replace the standard objective (i.e., the expectation of Q function over some initial state distribution) with the expectation of Q function over both the initial and an $\eta$-weighted future state distribution, where $\eta$ is some probability distribution over the support set of nonnegative integers. Built on this formulation, this paper proposes GIRL (and the resulting algorithm MEGAN), which follows the framework of GAIL to learn a policy that matches the $\eta$-weighted variant of occupancy measure of the expert policy. Experimental results on MuJoCo are provided to demonstrate the effectiveness of GIRL.

**Summary Of The Review:**

This paper proposes an extension of GAIL that addresses longer-term imitation. I like the insight provided by this paper, but the overall technical novelty is somewhat limited. The empirical results look promising, but there are several places that would require better explanations. As a result, I lean towards rejection in its current form.

---

### Decision · Program_Chairs · 2022-01-20

**Decision:**

Reject

**Comment:**

The paper extends the maximum entropy inverse reinforcement learning (IRL) framework by changing the optimal criterion used in reinforcement learning (RL). This novel criterion is an expectation of the Q-values over a weighted distribution over states and actions induced by a policy, which is in contrast to the standard criterion that is an expectation over the initial state distribution.

All the reviewers agree that the topic addressed in this paper is interesting and novel. On the other hand, there are some concerns about the technical novelty and relevance of the paper. Since the authors have not provided any feedback, the reviewers did not solve their concerns and they reach a consensus on rejecting this paper.